# Effectiveness of Prophylactic Human Cytomegalovirus Hyperimmunoglobulin in Preventing Cytomegalovirus Infection following Transplantation: A Systematic Review and Meta-Analysis

**DOI:** 10.3390/life12030361

**Published:** 2022-03-02

**Authors:** Markus J. Barten, Fausto Baldanti, Alexander Staus, Christian M. Hüber, Kyriaki Glynou, Andreas Zuckermann

**Affiliations:** 1Department of Cardiovascular Surgery, University Heart and Vascular Center Hamburg, 20246 Hamburg, Germany; 2Department of Clinical, Surgical, Diagnostic, and Pediatric Sciences, University of Pavia, 27100 Pavia, Italy; f.baldanti@smatteo.pv.it; 3Microbiology and Virology Unit, Fondazione IRCCS Policlinico San Matteo, 27100 Pavia, Italy; 4Biotest AG, 63303 Dreieich, Germany; alexander.staus@biotest.com (A.S.); christian.hueber@biotest.com (C.M.H.); kyriaki.glynou@biotest.com (K.G.); 5Division of Cardiac Surgery, Medical University of Vienna, 1090 Vienna, Austria; andreas.zuckermann@meduniwien.ac.at

**Keywords:** cytomegalovirus, CMV infection, human CMV hyperimmunoglobulin, CMVIG, prophylaxis, transplantation

## Abstract

Cytomegalovirus (CMV) is a common infection occurring in patients undergoing solid organ transplantation (SOT) or hematopoietic stem cell transplantation (HSCT). CMV-specific hyperimmunoglobulin (CMVIG) has been used for the past four decades and is typically administered either prophylactically or pre-emptively. The present meta-analysis evaluated CMV infection rates in SOT patients who received prophylactic CMVIG. PubMed and the Cochrane Library were searched for studies published up to October 2021. The primary endpoint was CMV infection rate. Thirty-two SOT studies were identified (n = 1521 CMVIG-treated and n = 1196 controls). Prophylactic CMVIG treatment was often associated with a lower risk of CMV infection in transplant recipients. The average CMV infection rate was 35.8% (95% confidence interval [CI]: 33.4–38.2%) in patients treated prophylactically with CMVIG and 41.4% (95% CI: 38.6–44.2%) in the control group not receiving CMVIG (p = 0.003). Similar results were observed in analyses limited to publications evaluating currently available CMVIG products (Cytotect CP and Cytogam; p < 0.001). In combination with the established safety profile for CMVIG, these results suggest that prophylactic CMVIG treatment in patients undergoing solid organ transplantation may be beneficial, particularly in those at high risk of CMV infection or disease.

## 1. Introduction

Cytomegalovirus (CMV) is a common opportunistic pathogen of the *Herpesviridae* family, with an estimated mean global seroprevalence of 83% in the general population [1]. Seronegative individuals may experience a primary infection followed by a long period of latency [2]. Following the primary infection, CMV may reactivate or a new strain of CMV may infect the individual. Both infection and reactivation cause minimal or no symptoms in most immunocompetent people, but can lead to uncontrolled viral replication and serious illness in immunocompromised patients. Notably, viremia and viral dissemination to multiple organs can result in end-organ CMV disease, such as pneumonitis, hepatitis, retinitis, mononucleosis, or gastroenteritis [1,3,4].

CMV infection can be a serious complication in patients receiving transplants, including both solid organ transplantation (SOT), such as lung, heart, liver, and kidney transplantation, and hematopoietic stem cell transplantation (HSCT) [5,6,7]. In addition to the direct effects of CMV infection described above, CMV is also associated with indirect immunomodulatory effects in transplant recipients, such as graft rejection, atherosclerosis, and secondary opportunistic infections, leading to increased mortality in these patients [3,8,9]. The risk of CMV infection and disease depends on the serostatus of the recipient and donor. In solid organ transplantation, seronegative recipients receiving organs from seropositive donors (D+/R−) are at the highest risk due to the likelihood of primary infection [10]. Seropositive recipients with donors who are either seropositive or seronegative (D+/R+ or D−/R+) are at moderate risk of CMV disease due to CMV reactivation or reinfection. Seronegative recipients of organs from seronegative donors (D−/R−) are generally at low risk. CMV infection and disease rates can vary further based on the type of SOT. For instance, in the absence of prophylaxis, rates of CMV disease in D+/R− patients ranged from 50% to 91% in lung and lung–heart transplants, from 29% to 74% in heart transplants, and from 45% to 65% in liver and in kidney and/or pancreas transplants [10]. Hypogammaglobulinemia after solid organ transplantation also confers an increased risk of CMV infection [11].

Initiating strategies to prevent CMV infection or reactivation is recommended for patients undergoing transplantation [12]. Preventive treatment against CMV after transplantation may be administered either prophylactically, with antiviral treatment initiated immediately after transplantation, or “pre-emptively,”, employing assays to detect the virus and initiating treatment only when the viral load has reached a defined threshold in the blood [12,13,14,15]. The use of antiviral drugs has strongly reduced CMV-related morbidity and mortality in transplant recipients. However, they are associated with significant toxicity and may become ineffective in cases of resistant or refractory CMV infection. Antiviral-resistant CMV is an uncommon but important issue associated with accrued morbidity and mortality, notably in SOT recipients [15]. Alternative and complementary CMV prevention strategies exist. Among them, CMV-specific hyperimmunoglobulin (CMVIG) has been employed to reduce the risk of CMV infection and associated complications for four decades, either alone—mainly in the preantiviral era—or as a combination therapy with virostatics for CMV prophylaxis or treatment [16,17,18]. Human CMVIG acts notably by binding to the viral surface, thereby neutralizing the ability of the virus to enter host cells, and by interacting with immune cells to mediate antibody-dependent cellular cytotoxicity and other complex immunomodulatory effects [16]. The selection of therapy depends on a number of patient characteristics, including the patient’s degree of immunosuppression and the patient’s serology status, along with the serology status of the donor [12]. Renal insufficiency, which limits the dose of ganciclovir/valganciclovir that can be used due to cytotoxicity, and the potential for bone marrow depression with immunosuppressant and valganciclovir use, also factor into treatment decisions [12,19].

Several studies suggest the benefit of prophylactic human CMVIG on the clinical outcome of transplant recipients [20]. We undertook meta-analyses to evaluate the efficacy of human CMVIG as a prophylactic treatment in monotherapy or in combination with antivirals, to prevent CMV infection in SOT and HSCT patients. We identified 36 eligible studies, 32 in SOT patients and 4 in HSCT patients. To limit heterogeneity, we therefore focused our meta-analyses on the 32 SOT studies. The first meta-analysis assessed the efficacy of all human CMVIG formulations employed in eligible SOT studies, while a second meta-analysis assessed the efficacy of the two currently marketed CMVIG products (Cytotect CP and Cytogam). In both analyses, prophylactic CMVIG treatment was associated significantly with a lower risk of CMV infection in SOT patients, suggesting that prophylactic CMVIG may provide a clinical benefit after solid organ transplantation.

## 2. Materials and Methods

### 2.1. Literature Search

A literature search of the PubMed and the Cochrane library databases was performed to identify reports published until October 2021, using the following search terms: Cytotect AND transplant, Cytogam AND transplant, or CMVIG AND transplant. Citations of retrieved publications were also screened for additional eligible studies.

### 2.2. Eligibility Criteria

The inclusion criteria for meta-analyses allowed studies in which CMVIG was given as a prophylactic agent at the time of transplantation, and in which the results included the CMV infection rate as an endpoint. Studies were included whether they were controlled with no prophylaxis or a non-CMVIG prophylactic treatment (including with historical controls) or were uncontrolled, whether they were prospective or retrospective, and whether they were observational or randomized. Studies employing any CMVIG product were included in the first set of meta-analyses, whereas the second set of meta-analyses specifically addressed the two currently available CMVIG products: (i) Cytotect CP (and its predecessor Cytotect) and the other brand names under which it is marketed (e.g., Megalotect; Biotest AG, Dreieich, Germany)***,*** and (ii) Cytogam (Saol Therapeutics, Roswell, GA, USA; formerly manufactured by CSL Behring AG, Bern, Switzerland). Studies that did not fully report the frequency of CMV infection were excluded, as were case reports and publications that reported only end-organ CMV disease rates such as CMV pneumonia or CMV retinitis. Publications that reported outcomes for fewer than five patients in total were also excluded from the meta-analysis. Only full-text articles in English or German were considered for further assessment.

It should be noted that while the studies ranged in their publication date from 1986 to 2020, the beta-propiolactone treatment was removed from the Cytotect production process in 2013 (with the introduction of Cytotect CP), yielding a potentially more potent product; none of the meta-analyses distinguished between outcomes with these two differing Cytotect formulations [21].

### 2.3. Literature Screening and Data Extraction

Two reviewers independently screened titles and abstracts according to the exclusion and inclusion criteria. The selected full-text articles were further assessed for eligibility by both reviewers according to the exclusion and inclusion criteria. Disagreements were discussed and solved between reviewers. Study data were extracted using a standardized data sheet.

### 2.4. Primary Endpoint

The primary endpoint was the rate of CMV infection. Where possible, CMV infection was defined as CMV DNAemia or viremia without symptoms. Otherwise, the definition of CMV infection chosen by the authors of that publication was used as the endpoint. Some such definitions, particularly in older publications, required the detection of the CMV pp65 antigen (pp65 antigenemia) or anti-CMV antibodies in serum not attributable to immunoglobulin infusion (e.g., antibody seroconversion defined by either the appearance of IgM or a specified increase in IgG titer).

### 2.5. Additional Outcomes

Although this study was not designed to evaluate additional outcomes, such as time to infection, rejection rate, and adverse outcomes associated with CMVIG, these data were collected where available and were summarized descriptively.

### 2.6. Statistical Analysis

The number of patients in each treatment group in each study was recorded, as was the frequency of CMV infection for each treatment group, with the relative frequency of CMV infection assessed using the Clopper–Pearson method to calculate exact 2-sided 95% confidence intervals (CIs) based on binomial distributions. The Clopper–Pearson method is considered to be a conservative measurement [22]. Reports of the absolute frequency of CMV infection were taken directly from the original publications. Comparisons of the frequency of CMV infection in the CMVIG groups and the control groups employed a 2-sided chi-square test. All statistical analyses were performed using SAS Version 9.4, except for heterogeneity and publication bias testing. The latter analyses were performed with R Version 4.1.1 (package “meta”), considering studies with both treatment arms (CMVIG and control). Heterogeneity was evaluated using the I^2^ index. If the I^2^ index was between 50% and 75%, heterogeneity was evaluated as moderate. If the I^2^ index was >75%, heterogeneity was evaluated as considerable. Potential publication bias was assessed using funnel plots. Funnel plots’ asymmetry was evaluated using the Egger’s test [23].

## 3. Results

### 3.1. Search Results

Of the 186 unique records that were screened, 58 full-text publications were evaluated for eligibility, and 36 studies meeting the inclusion criteria were identified (Figure 1) [24,25,26,27,28,29,30,31,32,33,34,35,36,37,38,39,40,41,42,43,44,45,46,47,48,49,50,51,52,53,54,55,56,57,58,59]. Out of the 36 eligible studies, 32 were conducted in SOT patients and 4 in HSCT patients. Due to major differences in clinical settings and in patient management between SOT and HSCT, and to avoid introducing heterogeneity bias into the analysis, the four HSCT studies were excluded from the meta-analysis (Figure 1). Thus, a total of 32 SOT studies were included in this meta-analysis (Figure 1) [24,25,26,28,29,30,31,32,33,34,35,36,37,38,40,41,42,43,44,46,47,48,49,50,51,52,53,54,55,56,57,58].

The characteristics of the studies included in all meta-analyses are shown in Appendix A. A total of 19 studies used prospective data and 13 used retrospective data. In total, 13 of the 32 studies involved patients undergoing kidney transplants, 6 studies involved heart transplants, 6 focused on lung transplants, 5 addressed liver transplants, 1 assessed kidney and heart transplants, and 1 examined pancreas and kidney transplants.

The SOT group generally thought to be at the highest risk for CMV infection is CMV seronegative recipients with seropositive donors (D+/R−) [10,12]. However, the serological status of patients evaluated varied among studies: (i) 10 studies included only seronegative recipients with seropositive donors; (ii) 17 studies included a mix of serostatus combinations of recipients and donors (though 4 of these studies did not include seronegative recipients with seronegative donors, a group with a relatively low risk of CMV infection); (iii) 3 studies did not report the serostatus of donors; and (iv) 2 studies did not report the serostatus of donors or recipients. Four studies assigned patients with higher-risk serostatus to the CMVIG group and assigned those with lower risk to the control antiviral prophylaxis group.

In total, 26 of the 32 identified studies included at least 1 study arm, in which patients received Cytotect or Cytogam as monotherapy or in combination with another agent. A total of 11 of these 26 studies were in patients undergoing kidney transplants, 6 in heart transplant patients, 3 in lung transplants, 4 in liver transplants, 1 in kidney and heart transplants, and 1 in pancreas and kidney transplants.

Approaches to immunosuppression regimens varied to some extent but typically involved the calcineurin inhibitor (CNI) cyclosporine, steroids, and azathioprine (Appendix A). Other agents used included the CNI tacrolimus, the monoclonal antibodies OKT3 and basiliximab, the inosine monophosphate dehydrogenase inhibitor mycophenolate mofetil (MMF), the anti-lymphocyte antibody preparations anti-lymphocyte globulin (ALG) and anti-thymocyte globulin (ATG), and the mammalian target of rapamycin inhibitor sirolimus.

The most frequently used prophylactic CMVIG regimen involved the administration of 150 mg/kg within 72 h of transplantation and every 2 weeks thereafter. Dosing in other cases usually ranged from 100 mg/kg to 150 mg/kg or from 1 to 2 mL/kg, with the schedule of dosing also being highly variable. CMVIG prophylaxis was most often continued for 3–4 months post-transplantation, although the duration varied from 1 month to 1 year. In nine studies, some patients also received ganciclovir in combination with CMVIG.

Heterogeneity between included studies was moderate, with an I^2^ index of 60%, considering all transplant studies, and of 57%, considering studies limited to the use of Cytotect or Cytogam. No publication bias was evident in all performed analyses, based on funnel plots’ evaluation (Egger’s test *p* > 0.05) (Appendix A).

### 3.2. All-CMVIG Meta-Analysis

In the 32 SOT studies included in the all-CMVIG analysis, study population sizes ranged from as few as 3 patients to as many as 377 patients in the CMVIG group; the total patient population was 1521 in the pooled CMVIG group and 1196 in the pooled control group (Figure 2).

Of the 25 studies with a control group, 15 had lower rates of CMV infection with CMVIG than controls, 1 had the same rate with CMVIG and controls, and 9 had higher rates with CMVIG than controls. The average rate of CMV infection was 35.8% in the pooled CMVIG group and 41.4% in the pooled control group (*p* = 0.003). The 2-sided Clopper–Pearson 95% CI was 33.4–38.2% in the pooled CMVIG group and 38.6–44.2% in the pooled control group (Table 1).

Ten studies included SOT patients in the highest-risk group (D+/R−). Infection rates reported in these 10 studies are shown in Appendix A. Of the seven D+/R− studies with a control group included in the analysis, five had lower rates of CMV infection with CMVIG than controls, one had the same rate with CMVIG and controls, and one had a higher rate with CMVIG than controls.

### 3.3. Meta-Analysis of Cytotect and Cytogam

Twenty-six studies were included in this analysis, which was limited to SOT patients receiving Cytotect or Cytogam for prophylactic CMVIG (Figure 3). A total of 1350 patients were included in the pooled Cytotect/Cytogam group and 833 in the pooled control group.

Of the 19 studies with a control group, 12 had lower rates of CMV infection with Cytotect or Cytogam than controls, 1 had the same rate with Cytotect or Cytogam and controls, and 6 had higher rates with Cytotect or Cytogam than controls. The average rate of CMV infection was 35.9% in patients receiving Cytotect/Cytogam versus 44.4% in the pooled control group (*p* < 0.001). The 2-sided Clopper–Pearson 95% CI was 33.4–38.6% in the pooled Cytotect/Cytogam group and 41.0–47.9% in the pooled control group (Table 1).

### 3.4. Meta-Analysis of Recent Studies in the Era of Modern CMV Diagnosis

To evaluate the potential benefit of CMVIG in the era of modern CMV diagnosis and management, a sub-analysis was conducted focusing on studies detecting CMV infections by pp65 antigenemia and/or CMV DNAemia (using quantitative polymerase chain reaction). This sub-analysis included five SOT studies covering the period from 2005–2020. Heterogeneity between included studies was slightly higher than in the previous analyses but remained moderate, with an I^2^ index of 72%. No publication bias was detected in this analysis (funnel plot, Egger’s test *p* > 0.05) (Appendix A). The total patient population was 182 in the pooled CMVIG group and 288 in the pooled control group (Figure 4).

The five SOT studies in this sub-analysis included a control group. Of these, four had lower rates of CMV infection with CMVIG than controls, and one had a higher rate of CMV infection with CMVIG than controls. The average rate (95% CI) of CMV infection was 36.3% (29.3–43.7%) in the pooled CMVIG group and 39.9% (34.2–45.8%) in the pooled control group (*p* = 0.426) (Table 1).

### 3.5. Additional Outcomes

The timing of infections was reported for 18 of the 32 studies (Appendix A), though reporting was heterogeneous, with some studies giving time to infection and others providing the percentage of patients with infections in a given time period. No differences between CMVIG- and non-CMVIG-treated groups were apparent.

In total, 16 of 32 studies made a statement on potential adverse events associated with CMVIG (Appendix A), with 8 of the 16 studies stating that no adverse events (or no adverse events requiring discontinuation) occurred. In the eight studies reporting adverse events, flushing, back and/or muscle pain, and rashes were the most common events.

## 4. Discussion

Hyperimmunoglobulins have been effectively employed to reduce the risk of CMV infection and its associated complications for four decades, and yet there have been few recent analyses on the efficacy of CMVIG in the prevention of CMV infection that take into account the full time period in which CMVIG has been used in the clinical setting. We undertook meta-analyses of all available forms of CMVIG, as well as currently available products, Cytotect CP and Cytogam, to determine CMV infection rates in SOT recipients. In both meta-analyses, treatment with CMVIG reduced the risk of CMV infection versus the study control. These differences in the pooled results were statistically significant.

Patient populations included in this analysis were not restricted based on their serological status, which may have introduced variability in the results, as the degree of CMV infection risk was higher in some studies than in others. Of note, in some SOT studies, CMV-seronegative recipients were assigned to the CMVIG group while seropositive patients were assigned to the control group [25,31,38,58]. This presumably biased the results toward a higher likelihood of CMV infection in the CMVIG-treated group than in the control group. Thus, the current meta-analysis may have underestimated the difference in infection rates between CMVIG-treated and non-CMVIG-treated patients.

The reductions in CMV infection with prophylactic CMVIG use that were identified here may translate to additional improved outcomes, although the data in this analysis were insufficient to evaluate additional outcomes due to differences in reporting across studies. A previous meta-analysis that only included randomized trials suggested that the prophylactic use of CMVIG has a beneficial effect on clinical outcomes, including total survival (rate ratio of death [95% CI]: 0.67 [0.47–0.95]) and the prevention of CMV-associated death (rate ratio [95% CI]: 0.45 [0.24–0.84]) in SOT recipients [20]. However, that analysis did not find a significant difference in the incidence of CMV infections. A large study of pediatric patients who underwent solitary primary heart transplantation in the United States determined that CMVIG prophylaxis with or without antivirals was associated with a reduction in graft loss and death compared with no prophylaxis [60]. That study did not evaluate CMV infection rates.

Human hyperimmunoglobulins are formulations prepared from plasma pools from large numbers of blood donors with elevated titers of antibodies to a given virus [18]. The primary function of CMVIG is to provide passive immunity by neutralizing circulating CMV particles and facilitating their elimination [17]. CMVIG also exerts enhancing and suppressive immunomodulatory functions that might contribute to the control of the direct and indirect effects of post-transplant CMV infection [16,17,18,61]. Immunocompromised patients do not produce the cellular and humoral immune responses that would typically defend against CMV infection or reactivation. Immunomodulatory actions proposed for human CMVIG include the suppression of functional dendritic cell maturation and the inhibition of T-cell proliferation, potentially leading to lower rates of transplant rejection [16]. Some reports have suggested that CMVIG may not only be useful as prophylaxis but also as rescue therapy upon the detection of CMV infection, but the current analysis focused on studies evaluating the labeled use in prophylaxis, in which CMVIG prophylaxis was started around the day of transplantation. It should also be noted that the specific criteria for the diagnosis of CMV infection varied among studies. This variability is in part due to the broad time period covered by the studies in this analysis, as older studies tended to rely on viral cultures, shell vial assays, or serological testing to detect CMV, while newer studies generally measured pp65 antigenemia and/or CMV DNAemia using polymerase chain reaction. A sub-analysis focusing on studies detecting CMV by pp65 antigenemia and/or CMV DNAemia only included five SOT studies (covering the period 2005–2020) and showed a trend toward a benefit of CMVIG (with four of five studies having a lower rate of CMV infection in the CMVIG group compared to the control group), albeit not statistically significant. Future studies will be required to confirm the benefit of CMVIG in the modern era of CMV prevention following transplantation.

Although this analysis was not designed to evaluate the safety of prophylactic CMVIG in transplant recipients, the studies included in this analysis that reported safety found CMVIG to be generally well-tolerated. Reported adverse events were consistent with the established safety profile for CMVIG [62,63], and rarely led to the discontinuation of treatment.

The present analysis has several limitations, mainly related to the heterogeneity of study designs, although the studies shared certain broad commonalities. Dosing of CMVIG also varied between studies, in part depending on which organ was being transplanted. Nearly every study used an immunosuppressive regimen of cyclosporine and steroids, while the use of azathioprine was almost as universal. The use of OKT3, ALG, ATG, and/or MMF was less common, though they were still employed for immunosuppression in many studies, whereas only one study used sirolimus and one study used basiliximab. It is also worth noting that the structure of immunosuppression protocols was inconsistent; the majority included just one stage of treatment, unless rejection occurred, whereas a minority of studies applied an induction regimen followed by a maintenance regimen. In addition, some immunosuppressants (e.g., everolimus and sirolimus) and antibody-induction therapies (e.g., ATG) have been shown to affect the risk of CMV infection [64,65,66,67,68], which may have contributed to some of the variation between studies. The immunosuppressive regimens in these studies may not reflect the regimens currently being used, limiting the generalizability of these results. Despite these limitations caused by study design heterogeneity, statistically significant differences in the rate of CMV complications were identified in association with the prophylactic use of CMVIG after transplantation.

In most of the studies analyzed, patients were not treated with CMVIG alone but were also administered concomitant agents that could be expected to affect outcomes, most commonly, the antiviral agent ganciclovir. This follows current consensus guidelines that note a potential benefit of CMVIG in combination with antivirals in thoracic organ transplant recipients with hypogammaglobulinemia [12].

While these studies were heterogeneous in design, they provide a global picture of prophylactic CMVIG use in real-world settings and allow for insights to be drawn from an aggregation of outcomes. The robustness of this analysis is supported by the consistently lower rates of CMV infection with CMVIG prophylaxis compared with controls in each meta-analysis. Additional studies are needed to confirm the benefit of prophylactic CMVIG treatments identified in this analysis, taking into account the current standard of care for transplant recipients. In that regard, real-world non-interventional studies assessing the diverse off-label use of CMVIG in routine clinical practice would provide important insights into the clinical benefits of CMVIG. Based on the outcome of this meta-analysis, an international multicenter prospective observational study in heart and lung transplant recipients will be initiated in the coming months of 2022. This study will explore the usage of CMVIG in real-world settings, i.e., in the context of other current CMV treatments and modern immunosuppressive regimens. This study will allow us to draw conclusions on the efficacy and safety of different prophylaxis and treatment approaches and evaluate their impact on the incidence of CMV infection, CMV disease, graft survival, and mortality.

## 5. Conclusions

The use of CMVIG in patients undergoing solid organ transplantation was shown in most studies to confer additional protection in preventing CMV infection compared with controls. CMVIG prophylaxis was also generally well-tolerated. These results suggest that patients undergoing lung, heart, kidney, or liver transplantation could be recommended for prophylactic CMVIG. CMVIG prophylaxis could be beneficial, especially to patients at high risk for CMV infection, such as D+/R− SOT patients, those who are highly immunosuppressed, and those who cannot receive antivirals due to antiviral-induced nephrotoxicity, neutropenia, or resistance.

## Figures and Tables

**Figure 1 life-12-00361-f001:**
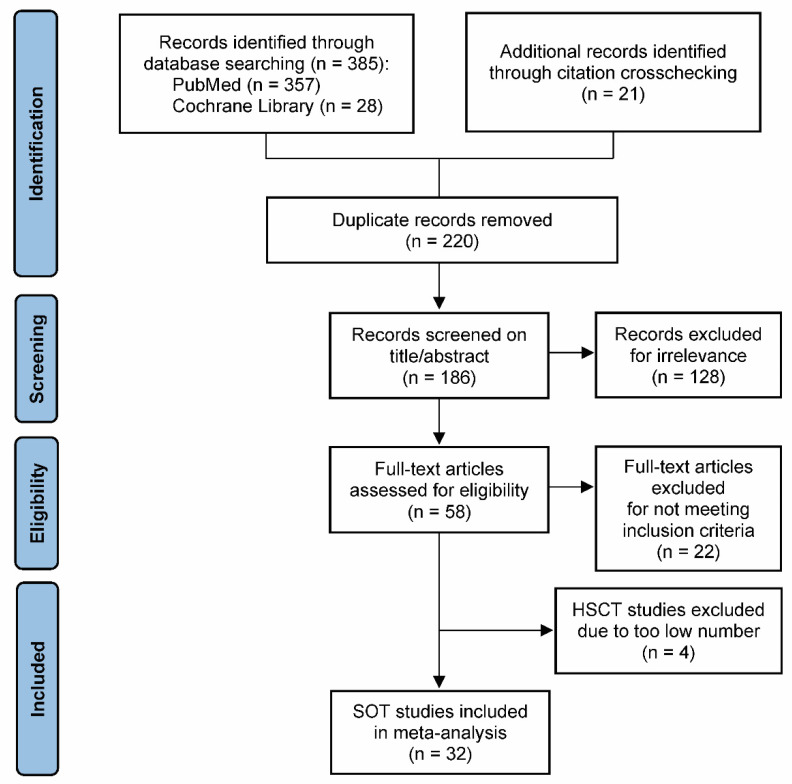
Flow diagram of study selection for the meta-analysis. Abbreviations: HSCT, hematopoietic stem cell transplantation; SOT, solid organ transplantation.

**Figure 2 life-12-00361-f002:**
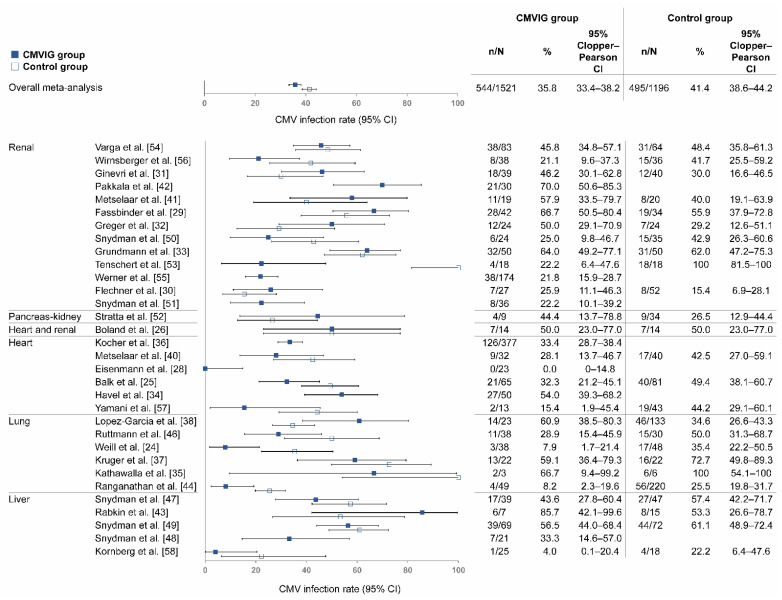
Rate of CMV infection among SOT patients receiving any prophylactic CMVIG. Abbreviations: CI, confidence interval; CMV, cytomegalovirus; CMVIG, cytomegalovirus-specific hyperimmunoglobulin; SOT, solid organ transplantation.

**Figure 3 life-12-00361-f003:**
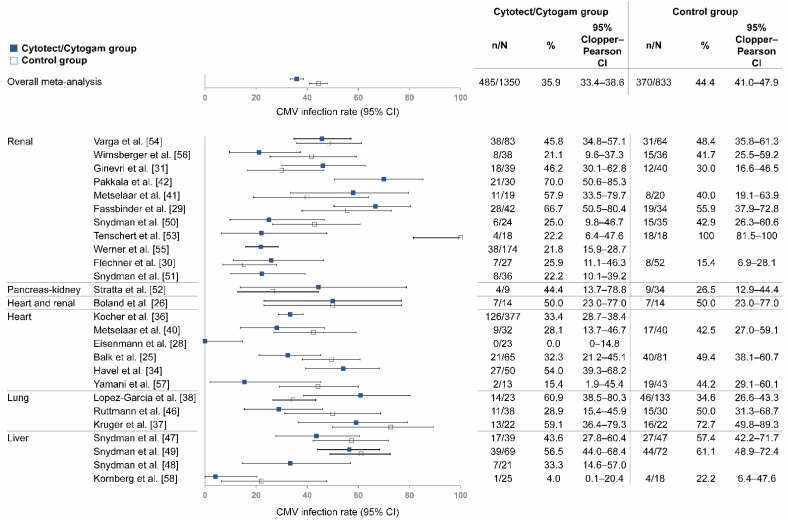
Rate of CMV infection among SOT patients receiving Cytotect or Cytogam. Abbreviations: CI, confidence interval; CMV, cytomegalovirus; CMVIG, cytomegalovirus-specific hyperimmunoglobulin; SOT, solid organ transplantation.

**Figure 4 life-12-00361-f004:**
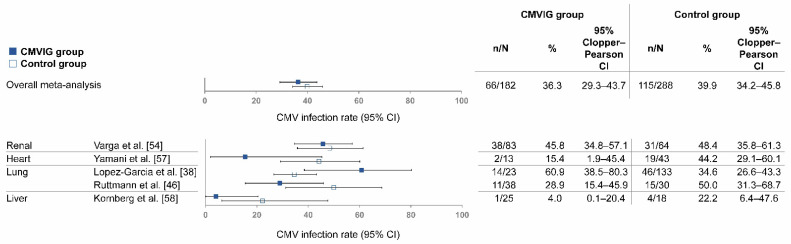
Rate of CMV infection among SOT patients receiving any prophylactic CMVIG, in studies detecting CMV infection via pp65 antigenemia and/or CMV DNAemia. Abbreviations: CI, confidence interval; CMV, cytomegalovirus; CMVIG, cytomegalovirus-specific hyperimmunoglobulin; SOT, solid organ transplantation.

**Table 1 life-12-00361-t001:** Meta-analysis of CMV infection rates in solid organ transplantation for CMVIG vs. control groups.

Prophylactic Treatment	CMVIG Group		Control Group		*p*-Value
	n/N	CMV Infection Rate, % (95% CI)	n/N	CMV Infection Rate, % (95% CI)	
Any prophylactic CMVIG	544/1521	35.8 (33.4–38.2)	495/1196	41.4 (38.6–44.2)	0.003
Cytotect/Cytogam	485/1350	35.9 (33.4–38.6)	370/833	44.4 (41.0–47.9)	<0.001
Any prophylactic CMVIG (Modern CMV diagnosis) ^1^	66/182	36.3 (29.3–43.7)	115/288	39.9 (34.2–45.8)	0.426

^1^ Defined as pp65 antigenemia and/or CMV DNAemia by quantitative PCR. Abbreviations: CI, confidence interval; CMV, cytomegalovirus; CMVIG, CMV-specific hyperimmunoglobulin.

## Data Availability

The data presented in this study are available within the article and Appendix A.

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
