# Peer review of "Effectiveness of Prophylactic Human Cytomegalovirus Hyperimmunoglobulin in Preventing Cytomegalovirus Infection following Transplantation: A Systematic Review and Meta-Analysis"

_life, 2022, doi:10.3390/life12030361_

Round 1
Reviewer 1 Report
Barten et al performed a meta-analysis to evaluate CMV infection rates in SOT patients who received prophylactic CMVIG. They found prophylactic CMVIG treatment was often associated with a lower risk of CMV infection in transplant recipients. They concluded that prophylactic CMVIG treatment in patients undergoing solid-organ transplantation might be beneficial, particularly in those at high risk of CMV infection.
Overall, this review article is well written. However, the role of CMVIG in prophylaxis might be limited in the era of antivirals, primarily due to limited effectiveness as a single agent compared to GCV or VGCV.
In this article, the authors have focused on “the ear of modern CMV diagnosis” as shown in Figure 4, but only 5 SOT studies were included. The average rate of CMV infection was 36.3% in the pooled CMVIG group and 39.9% in the pooled 246 control group (P = 0.426). Therefore, the effects of CMVIG in prophylaxis are not clear in the current situation. Furthermore, CMV “diseases” can be prevented by GCV or VGCV in most cases. It would be more important to analyze whether prophylactic CMVIG can prevent CMV disease or improve survival.
Author Response
We agree with the Reviewer that the analysis focusing on “the era of modern CMV diagnosis” is limited (5 SOT only) and that further studies are necessary to confirm the results presented in this manuscript. We also agree that investigating outcomes other that the CMV infection rate, such as CMV disease or mortality, are highly relevant and would be of interest. Unfortunately, the number of studies investigating these outcomes was too low to conduct a relevant meta-analysis. Future meta-analyses should also evaluate CMV disease and mortality as clinically relevant outcomes. A sentence was added in the revised manuscript in that regard (lines 381-384 in the “track changes” version).

Reviewer 2 Report
In this systematic review Barten and colleagues have analysed the effectiveness of prophylactic HCMV hyperimmunoglobulin (CMVIG) treatment in preventing CMV disease in transplantation. The authors compared 32 Solid organ transplant studies over the last 25 years and identified a significant decrease in CMV infection rates when CMVIG treatment was given. This is a well written and mostly clearly explained review, however the statistics section of the materials and methods omitted certain pertinent information including which statistical program was used (e.g., STATA or R – and if R if a particular package was utilised) and should possibly include a reference to the heterogeneity test used in this study. Another minor comment I have is with respect to figures 2, 3 and 4 which display the forest plots of the different studies compared in each meta-analysis, namely that the font is very small and hard to read, and I wondered if there was scope to increase the font a little.
Author Response
The statistics section was updated, now indicating the statistical programs used for the analyses. Heterogeneity testing and the interpretation of the I2 index are also described (lines 157-164 in the “track changes” version).
The font used in Figures 2, 3 and 4 was increased, as requested by the Reviewer.

Reviewer 3 Report
- Please indicate the data extraction, risk of bias and trial identification in the Material and Method section.
- Please provide the risk of bias assessment and indicate its impact and significance in the Results section.
- Please discuss the possible future research for human cytomegalovirus hyperimmunoglobulin in preventing cytomegalovirus infection following transplantation.
Author Response
- The Materials and Methods section was updated and restructured to better describe the method of literature search (section 2.1.), the eligibility criteria (section 2.2.), the method of literature screening and data extraction (section 2.3.), and the publication bias analysis (section 2.6, lines 159-164 in the “track changes” version).
- The Results section has been updated, now describing the results and significance of the publication bias evaluation for each analysis (lines 213-214 and 262-263 in the “track changes” version). Accordingly, a new Figure S1 was added to the Supplementary Materials, presenting the respective funnel plots and the results of the Egger’s test for funnel plots’ asymmetry (listed on lines 397-399 in the “track changes” version).
- The Discussion section was enriched by including possible future research for CMVIG in preventing CMV infection following transplantation, as recommended by the Reviewer (lines 374-384 in the “track changes” version). The conclusion was also amended (lines 390-394 in the “track changes” version). Notably, based on the outcome of this meta-analysis, some authors of the present manuscript are planning an international multicenter prospective observational study after heart and lung transplantation. This study will explore the usage of CMVIG in real-world settings, aiming to draw conclusions on the safety and efficacy of different prophylaxis and treatment approaches in routine clinical practice. Since this study is in the preparation phase and is not yet approved and registered, the aim of the planned study is briefly described in the discussion (lines 377-384 in the “track changes” version).

Round 2
Reviewer 3 Report
The manuscript has been significantly improved.